# Effect of diabetes on incidence of peritoneal dialysis-associated peritonitis

**Risa Ueda** **, Masatsugu Nakao\*, Yukio Maruyama, Akio Nakashima, Izumi Yamamoto, Nanae Matsuo, Yudo Tanno, Ichiro Ohkido, Masato Ikeda, Hiroyasu Yamamoto, Keitaro Yokoyama, Takashi Yokoo**

Division of Nephrology and Hypertension, Department of Internal Medicine, The Jikei University School of Medicine, Tokyo, Japan

* 2013nakaomasatsugu@gmail.com

## Abstract

### Background

Several reports on patients with diabetes mellitus (DM) treated by peritoneal dialysis (PD) have shown a higher risk of PD-associated peritonitis compared to non-DM (NDM) patients. The aim of this study was to investigate the incidence of PD-associated peritonitis in DM patients.

### Methods

We divided all patients who received PD at a single center between January 1980 and December 2012 into three groups according to era: Period 1 (n = 43, 1980–1993); Period 2 (n = 123, 1994–2004); and Period 3 (n = 207, 2005–2012). We investigated incidences of PD-associated peritonitis between patients with and without DM.

### Results

In Periods 1 and 2, incidence of PD-associated peritonitis was higher in the DM group than in the NDM group (P<0.05). However, no difference according to presence of DM was seen in Period 3. Multivariate Cox regression analysis revealed DM as a risk factor for incidence of PD-associated peritonitis in Periods 1 and 2, but not in Period 3 (hazard ratio [HR], 2.49; 95% confidence interval [CI], 1.15 to 5.23; HR, 2.36; 95%CI, 1.13 to 4.58; and HR, 0.82; 95%CI, 0.41 to 1.54, respectively). Furthermore, the peritonitis-free period was significantly shorter in the DM group than in the DM group in Periods 1 and 2, whereas no significant difference was seen in Period 3 (P<0.01, P<0.01 and P = 0.55, respectively). Moreover, a significant interaction was seen between diabetes and study period, and became less pronounced during Period 3(P<0.01).

### Conclusions

The increased risk of peritonitis in diabetics reported in previous periods has not been evident in recent years.

**Data Availability Statement:** All relevant data are within the manuscript.

**Funding:** YM has received scholarship funds from Baxter International Inc. and Terumo Corporation.

MI has received scholarship funds from Baxter International Inc. and Terumo Corporation. YT has received research grants from Baxter International Inc. (grant number 16CECPDAP0002). The specific roles of these authors are articulated in the 'author contributions' section. The funders had no direct involvement in the design or conduct of the study; collection, management, analysis, or interpretation of the data; or the preparation, review, or approval of the manuscript.

**Competing interests:** YM has received scholarship funds from Baxter International Inc. and Terumo Corporation. MI has received scholarship funds from Baxter International Inc. and Terumo Corporation. YT has received research grants from Baxter International Inc (grant number 16CECPDAP0002). This does not alter our adherence to PLOS ONE policies on sharing data and materials. Furthermore, there was nothing to declare with patent, product in development, and marked products. All other authors have no conflicts of interest to declare.

# Introduction

In a review from Japan reported at the end of 2013, a total of 43.8% of patients initiating dialysis were patients with diabetes [1], and the prevalence of diabetes mellitus (DM) has continued to increase as in other countries. Peritoneal dialysis (PD) is chosen less often by diabetic patients. Diabetic patients were less likely to receive PD as a first renal replacement therapy (RRT) than non-diabetic patients in North America (9.0% versus 10.1%) [2], Europe (14% versus 15%) [3], and Japan (4.9% versus 6.6% for diabetic patients commencing RRT) [4]. Clinicians are concerned that diabetic patients are likely to develop PD-associated peritonitis, given their immunocompromised state [5], and they mistake the process of PD as potentially contributing to visual disorder and peripheral neuropathy [6–8]. PD-associated peritonitis is an important complication related to both patient survival and technical survival. In addition, PD-associated peritonitis is a well-known risk factor for the development of encapsulating peritoneal sclerosis (EPS), one of the most serious complications [9]. Several studies have reported DM as a risk factor for PD-associated peritonitis [6–8, 10–14]. In contrast, DM was not a risk predictor in the Brazilian Peritoneal Dialysis Study (BRAZPD) [15].

We recently reported a 33-year, single-center cohort study including 527 PD patients and 377 episodes of PD-associated peritonitis [16]. In that study, the prevalence of PD-associated peritonitis declined dramatically over the course of 33 years, despite the increasing populations of both diabetic and aging patients. The chief causes of this change appear to have been several developments in medical technology, including the twin-bag system. Use of a twin-bag system has been reported to prevent PD-associated peritonitis by limiting the opportunities for contamination [17]. Given this situation, we presume that the impact of diabetes on the prevalence of PD-associated peritonitis has reduced.

The present post-hoc analysis tested the hypothesis that the prevalence of PD-associated peritonitis in diabetic patients did not differ from that in non-diabetic patients.

# Subjects and methods

## Study population

This study was a post-hoc study of the previously reported cohort study [16]. In the previous cohort, we retrospectively reviewed the medical records of all 527 patients who initiated PD at our hospital between January 1980 and December 2012. From this database, we excluded 154 of incident PD patients because of missing data on the history of PD-associated peritonitis or underlying diseases, leaving 373 patients eligible for this analysis. We then divided these 373 patients into three groups according to era: Period 1 (n = 43), PD initiated from 1980 to 1993; Period 2 (n = 123), PD initiated from 1994 to 2004; and Period 3 (n = 207), PD initiated from 2005 to 2012. In terms of PD devices and PD solutions, a single-bag method with conventional PD solution was mainly used in Period 1, a twin-bag system with conventional PD solution was mainly used in Period 2, and a twin-bag system with biocompatible PD solution was mainly used in Period 3. When patients used several devices and PD solutions, we allocated them to the group that they used for more than half of the duration of their PD therapy. In Japan, sterile systems came into use from 1993 to 1994. This term overlapped with Period 2. However, not every patient used a sterile system, unlike the twin bag system and biocompatible solution. Clinical information was obtained from medical records, but complete laboratory findings were not available in all cases because of the long study period, with data frequently missing from Period 1. The ethics committee at Jikei University Hospital approved this study protocol (approval number 30-295(9316)). Further, our ethics committee specifically waived the need for collection of patient consent.

## Diagnosis and treatment of PD-associated peritonitis

PD-associated peritonitis was diagnosed and patients recovering from peritonitis were identified using the criteria proposed by the International Society for Peritoneal Dialysis (ISPD). Patients were classified as having peritonitis if they satisfied at least two of the following criteria: i) presence of clinical symptoms (pain, fever, cloudy dialysate); ii) presence of >100 leukocytes/mm$^3$ of dialysate, with at least 50% polymorphonuclear neutrophils; or iii) positive results from culture or Gram stain. We defined 'recovery from PD-associated peritonitis' as recovery from the above-mentioned criteria. 'Duration of peritonitis' was defined the interval between onset of and recovery from PD-associated peritonitis. In addition, 'PD duration' was defined from the start of the PD therapy to study observation period.

## Statistical analysis

All data are presented as mean ± standard deviation or median and range, as appropriate. Values of P < 0.05 were considered statistically significant. Differences between groups were analyzed by Student's *t*-test or the Wilcoxon rank-sum test, as appropriate. Differences between the three groups were analyzed by one-way analysis of variance or the non-parametric Kruskal-Wallis test, as appropriate. Differences were considered statistically significant when the F value was less than 0.05. The Tukey-Kramer test was then used to determine the group that caused the difference. Nominal variables were tested using the chi-square test. The incidence rate of peritonitis was calculated by dividing the number of cases of incident peritonitis by the number of person-years of follow-up as the denominator under the Poisson assumption. Kaplan-Meier survival analysis was used to compare peritonitis-free times. The log-rank statistic was used to test differences between groups. Hazard ratios (HRs) and 95% confidence intervals (CIs) for the incidence of PD-associated peritonitis were assessed using Cox regression analysis and interaction analysis with the confounding factors of sex, age, study period and diabetes. Statistical analyses were performed using JMP for Windows version 10.0.2 (SAS Institute, Cary, NC).

## Results

### General

Table 1 details baseline characteristic of the 373 patients divided into the three groups, along with comparisons between diabetic and non-diabetic patients. DM was present in 92 patients. Mean age at PD initiation was 58±14 years in the DM group and 51±16 years in the non-DM (NDM) group (P<0.05). Age has been rising gradually in both the DM and NDM groups. The NDM group continued PD longer (49 months; range, 1–279 months) than the DM group (36 months; range, 0–119 months; P<0.05).

### Peritonitis incidence

A total of 268 episodes of peritonitis were identified. Incidence of PD-associated peritonitis (times per patient-year) was 0.16 in total, 0.21 for the DM group, and 0.15 for the NDM group (P<0.01). In Periods 1 and 2, the DM group showed higher incidences of PD-associated peritonitis than the NDM group (P<0.05). However, no difference according to the presence of diabetes was seen in Period 3.

In Period 1, the duration of peritonitis was 51.6±33.8 days. In Period 3, however, the duration of peritonitis was 20.8±15.1 days. Although length of treatment more than halved, the duration of peritonitis did not differ significantly between DM and NDM groups throughout the three periods.

**Table 1. Demographics and clinical characteristics.**

| | Whole period | | | Period 1 | | | Period 2 | | | Period 3 | | |
|---|---|---|---|---|---|---|---|---|---|---|---|---|
| | Total | DM | NDM | Total | DM | NDM | Total | DM | NDM | Total | DM | NDM |
| Number [n (%)] | 373 | 92 (24.7) | 281 (75.3) | 43 | 13 (30.2) | 30 (69.8) | 123 | 18 (14.6) | 105 (85.4) | 207 | 61 (29.5) | 146 (70.5) |
| total patient-year (year) | 1616.25 | 342.58 | 1273.66 | 288.08 | 58.83 | 226.25 | 713 | 114.66 | 598.33 | 618.16 | 169.08 | 449.08 |
| Male [n (%)] | 278 (74.5) | 78 (84.8) | 200 (71.2)† | 33 (76.7) | 12 (92.3) | 21 (70) | 87 (70.7) | 14 (77.8) | 73 (69.5) | 158 (76.3) | 52 (85.2) | 106 (72.6) |
| Age at PD initiation (years) | 53±15 | 58±14 | 51±16† | 45±13 | 51±12 | 42±13†† | 51±15 | 57±10 | 49±15† | 57±15 | 60±14 | 55±15†† |
| PD duration (months) | 45 (0–279) | 36 (0–119) | 49 (1–279)† | 101 (10–184) | 42 (12–112) | 111 (10–184)† | 72 (0–266) | 48 (3–119) | 73 (0–266)†† | 44 (0–279) | 29 (0–96) | 48 (1–279) |
| Peritonitis (total number of episodes) | 268 | 65 | 203 | 89 | 28 | 61 | 105 | 21 | 84 | 74 | 16 | 58 |
| Incidence of peritonitis (/patient-years) | 0.16 | 0.21 | 0.15†† | 0.31 | 0.49 | 0.27† | 0.15 | 0.27 | 0.13† | 0.12 | 0.09 | 0.13 |
| Duration of peritonitis | 34.7±28.8 | 34.4±27.9 | 34.8±29.2 | 51.6±33.8 | 54.3±29.3 | 49.5±35.9 | 35.6±28.6 | 27.5±21.1 | 37.8±30.1 | 20.8±15.1 | 18.3±15.8 | 21.6±15.0 |

Abbreviations: PD, peritoneal dialysis; DM, diabetes mellitus; NDM, non-diabetes mellitus.

† p<0.05 (comparison between DM and NDM groups)

†† p<0.01 (comparison between DM and NDM groups

We compared the development of peritonitis once to several times between DM and NDM groups for the 3 periods. No significant association was seen between groups in any period (Period 1 P = 0.27, Period 2 p = 1.00, Period 3 p = 0.51).

## Analysis of risk factor for peritonitis

Kaplan-Meier analysis was used to compare the peritonitis-free period between DM and NDM groups (Fig 1). The peritonitis-free period was significantly shorter in the DM group than in the NDM group in Periods 1 and 2, whereas no significant difference was seen in Period 3 (P<0.01, P<0.01 and P = 0.55, respectively).

In a Cox analysis including gender, age, and DM was an independent predictor for incidence of PD-associated peritonitis in Periods 1 and 2 (Period 1: HR, 2.49; 95%CI, 1.15 to 5.23; Period 2: HR, 2.36; 95% CI, 1.13 to 4.58). However, no correlation was seen between DM and PD-associated peritonitis in Period 3 (HR, 0.82; 95%CI, 0.41 to 1.54; Table 2).

Fig 2 shows the effects of diabetes and each clinical parameter on the incidence of PD-associated peritonitis. A significant interaction between diabetes and study period, and the influence of the presence of diabetes on the incidence of PD-associated peritonitis became less pronounced during Period 3.

## Causative microorganisms of PD-associated peritonitis in three terms

Table 3 shows causative microorganisms of PD-associated peritonitis in three terms. *Staphylococcus* species were significantly predominant in the DM group. In Period 1, *Staphylococcus* species infection was the most common in both groups. Although the difference was not significant, *Pseudomonas aeruginosa* was predominant in the NDM group. In Period 2, *Staphylococcus* species were the most common in both groups. One-third of patients (33%) showed culture-negative peritonitis episodes. Three (15%) diabetic patients displayed infections with methicillin-resistant *Staphylococcus epidermidis* (MRSE). After 2005, *Streptococcus* species increased, with eight (34.8%) diabetic patients showing *Streptococcal* peritonitis. For non-

Period 1

Period 2

Period 3

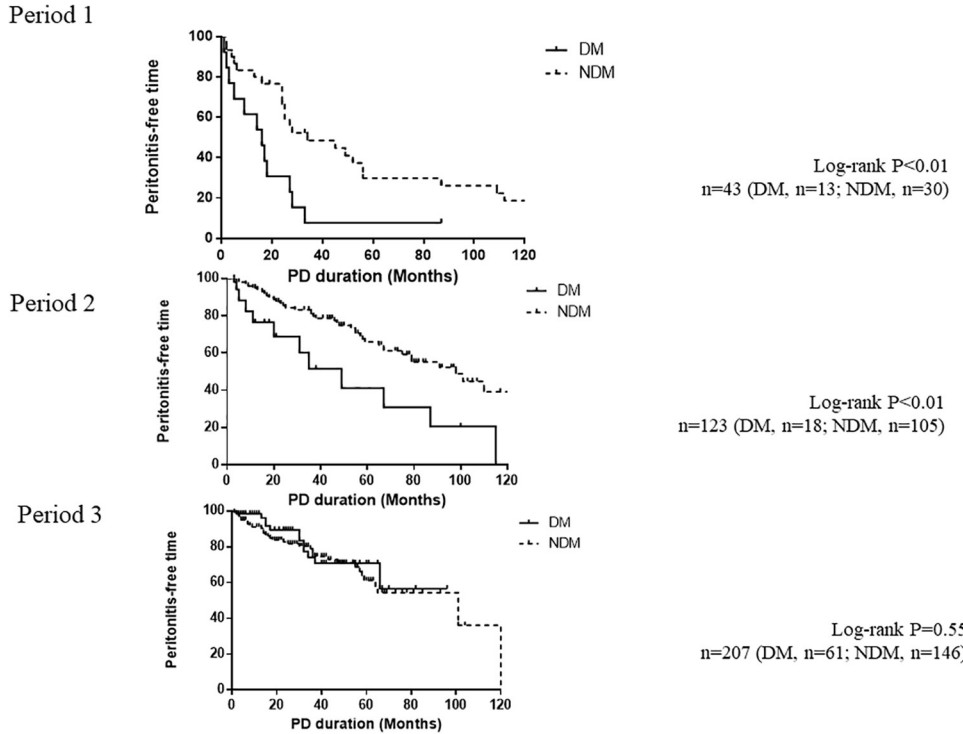

**Fig 1. Kaplan-Meier curves of peritonitis-free time.** Peritonitis-free time was compared between DM (solid line) and NDM (dashed line) groups. PD, peritoneal dialysis; DM, diabetes mellitus; NDM, non-diabetes mellitus.

diabetic patients, culture-negative peritonitis was the most frequent. No significant difference was seen between DM and NDM groups, but *Staphylococcus* species decreased, while *Streptococcus* species increased over the three periods.

## Discussion

This large, single-center cohort study evaluated 33 years of experience in the form of 268 episodes of PD-associated peritonitis. We divided patients into three groups by the use of PD

**Table 2. Hazard ratio for first peritonitis episode.**

|  | Unadjusted HR (95%CI) | Adjusted HR (95%CI) |
|---|---|---|
| Period 1 |  |  |
| Age (years) | 1.01 (0.98 to 1.04) | 1.00 (0.97 to 1.03) |
| Male sex | 1.24 (0.58 to 2.96) | 1.19 (0.51 to 3.02) |
| DM | 2.52 (1.18 to 5.16) | 2.49 (1.15 to 5.23) |
| Period 2 |  |  |
| Age (years) | 1.01 (0.99 to 1.03) | 1.00 (0.98 to 1.03) |
| Male sex | 1.18 (0.66 to 2.26) | 1.04 (0.57 to 2.01) |
| DM | 2.44 (1.19 to 4.60) | 2.36 (1.13 to 4.58) |
| Period 3 |  |  |
| Age (years) | 1.02 (0.99 to 1.04) | 1.02 (0.99 to 1.04) |
| Male sex | 0.93 (0.49 to 1.93) | 0.85 (0.45 to 1.78) |
| DM | 0.82 (0.41 to 1.53) | 0.82 (0.41 to 1.54) |

Abbreviations: HR, hazard ratio; CI, confidence interval; DM, diabetes mellitus.

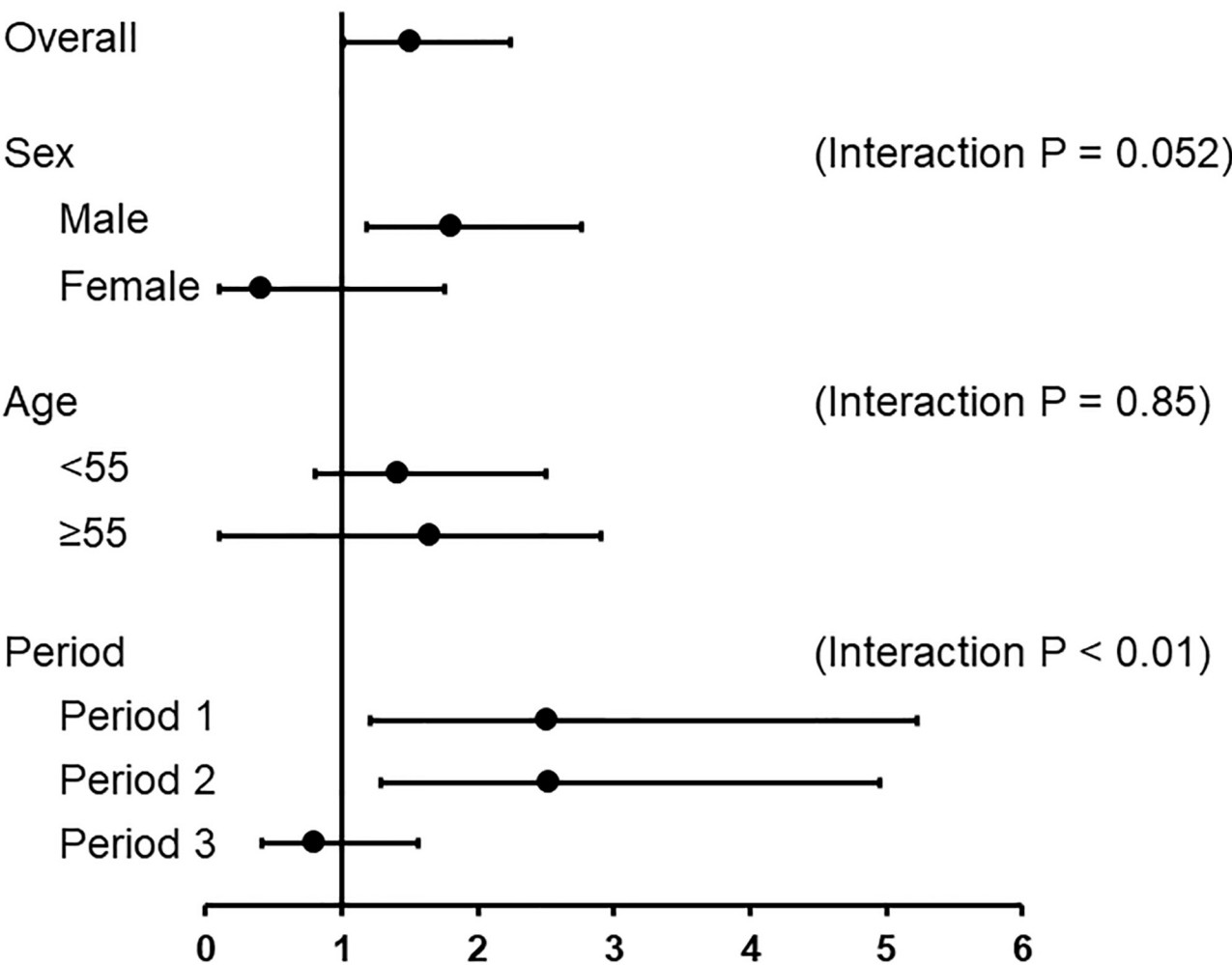

**Fig 2. Effect of the presence of diabetes and each clinical parameter on the incidence of PD-associated peritonitis.** CI, confidence interval.

devices and solutions use. In Japan, the twin-bag system was released from 1992 to 1993 and biocompatible solutions were released from 2000 to 2004. Our period classification corresponded to that used in a paper from Germany by Kitterer et al [18]. That study analyzed 351 adult patients with peritonitis in three time periods from 1979 to 2014.

Before 2004, the incidence of PD-associated peritonitis was higher and the peritonitis-free period was shorter in the DM group than in the NDM group, but no such differences were seen after 2005. In addition, a significant interaction was seen between diabetes and study period, and the influence of the presence of diabetes on the incidence of PD-associated peritonitis became less pronounced during Period 3. This means that the increased risk of peritonitis in diabetics observed in previous periods has not been seen in recent years.

Our study demonstrated that *Staphylococcus species* were predominant in DM. However, *Staphylococcus aureus, Staphylococcus epidermidis, methicillin-resistant S. aureus (MRSA), MRSE,* and other *Staphylococcus species* were not associated with DM. This result was similar to Australia and New Zealand Dialysis and Transplant Registry (ANZDATA) study [19].

**Table 3. Spectrum of PD-peritonitis episodes over the 33 years.**

| Causative microorganisms n(%) | Total | | | Period 1 | | | Period 2 | | | Period 3 | | |
|---|---|---|---|---|---|---|---|---|---|---|---|---|
| | DM | non DM | P | DM | non DM | P | DM | non DM | P | DM | nonDM | P |
| *Total* | 69 | 129 | | 26 | 17 | | 20 | 67 | | 23 | 45 | |
| *Staphylococcu* sp. | 28 | 34 | 0.04 | 15 | 8 | 0.49 | 8 | 18 | 0.26 | 5 | 8 | 0.69 |
| *Staphylococcus aureus* | 11 | 12 | | 10 | 5 | | 0 | 5 | | 1 | 2 | |
| MRSA | 1 | 2 | | 0 | 0 | | 1 | 1 | | 0 | 1 | |
| *Staphylococcus epidermidis* | 10 | 11 | | 5 | 2 | | 2 | 5 | | 3 | 4 | |
| MRSE | 3 | 3 | | 0 | 0 | | 3 | 3 | | 0 | 0 | |
| Other *Staphylococcus* sp | 3 | 6 | | 0 | 1 | | 2 | 4 | | 1 | 1 | |
| *Streptococcus* sp. | 12 | 21 | 0.96 | 3 | 1 | 0.63 | 1 | 9 | 0.44 | 8 | 11 | 0.46 |
| *Enterococcus* sp. | 1 | 6 | 0.41 | 0 | 0 | | 1 | 6 | 0.67 | 0 | 0 | |
| Other GPC | 3 | 0 | 0.08 | 0 | 0 | | 1 | 0 | 0.25 | 2 | 0 | 0.1 |
| GPR | 0 | 1 | 0.77 | 0 | 0 | | 0 | 1 | 0.56 | 0 | 0 | |
| *Pseudomonas aeruginosa* | 5 | 13 | 0.61 | 4 | 7 | 0.09 | 0 | 3 | 0.57 | 1 | 3 | 0.76 |
| *Esherichia coli* | 0 | 4 | 0.32 | 0 | 0 | | 0 | 4 | 0.57 | 0 | 0 | |
| Other GNR | 5 | 5 | 0.53 | 0 | 0 | | 2 | 2 | 0.26 | 3 | 3 | 0.39 |
| Fungi | 1 | 2 | 0.61 | 1 | 0 | 0.8 | 0 | 2 | 0.4 | 0 | 0 | |
| NTM | 0 | 1 | 0.77 | 0 | 0 | | 0 | 0 | | 0 | 1 | 0.71 |
| Polymicrobial infection | 1 | 6 | 0.41 | 0 | 0 | | 0 | 0 | | 1 | 6 | 0.41 |
| Negative | 13 | 36 | 0.15 | 3 | 1 | 0.63 | 7 | 22 | 0.89 | 3 | 13 | 0.22 |

As the time has progressed, infections involving *Staphylococcus species* have decreased, while those with *Streptococcus species* have increased. *Streptococcal* peritonitis has been demonstrated in 5–11.7% of cases in most studies [20,21]. The paper by Shukla *et al.* observed 104 cases of *Streptococcal* peritonitis in 68 patients over a period of 10 years [21]. They stated that the rate of *Streptococcal* peritonitis was increased, and viewed that the decline of *Staphylococcus* peritonitis relatively increased *Streptococcus* infections with Y-set systems and routine exit site care.

Alhough the ISPD has stated that frequencies of culture-negative peritonitis should be less than 15%, our study showed in Period 2 and for the NDM group in Period 3, culture-negative peritonitis accounted for more than 15% of cases [22].

According to other reviews of the literature examining the relationship between DM and PD-associated peritonitis, several studies reported DM as a risk for PD-associated peritonitis [6–8, 10–14]. Diabetic patients are compromised hosts and experience many complications. Furthermore, diabetic patients mistake the process of PD as potentially contributing to visual disorder and peripheral neuropathy [6–8]. Joshi et al. found that glucose load impaired the peritoneal defense system [23]. The presence of DM may affect the incidence of PD-associated peritonitis via several mechanisms.

The ISPD has recommended teaching PD patients and caregivers about appropriate catheter exit care and the use of prophylactic antibiotics [22]. Although we unfortunately did not have specific data to analyze, we tried to observe ISPD guidelines. Patients and caregivers have been taught techniques for PD on initiation of PD and became PD-associated peritonitis. For example, they were taught aseptic technique for connection, care for exit site, recognition about infection, and timing of medical examination. Instruction was provided on disinfection the catheter exit site with povidone iodine. The attending doctor examined catheter exit site for each patient every month. Systemic antibiotics were administered prophylactically prior to PD catheter insertion. We used intravenous cefazolin unless clinically contraindicaed.

Improvement of glycemic control is one factor that may help reduce the risk of PD-associated peritonitis. DPP-4 inhibitors and GLP-1 agonists currently play central roles in the treatment of dialysis patients with DM. These drugs were approved between 2009 and 2010 in Japan. The use of DPP-4 inhibitors was found to significantly improve hemoglobin (Hb)A1c levels and hyperglycemia in patients receiving PD [24,25]. Furthermore, GLP-1 agonists significantly decreased average blood glucose levels among diabetic patients undergoing PD [26]. Although we did not show data on glycemic control, diabetic patients might have benefitted from these treatments in Period 3. According to a report from Spain, the time to first episode of peritonitis did not differ between patients with HbA1C values ≤7.1% and those with values >7.1% [27]. In contrast, Lee et al. indicated that better patient survival with PD was influenced by the degree of glycemic control [28].

The mechanisms by which biocompatible PD solutions help prevent PD-associated peritonitis have not been established. Several reports have suggested that biocompatible PD solutions suppress peritoneal fibrosis and decrease the incidence of PD-associated peritonitis [29]. We have previously reported that biocompatible PD solutions did not exert favorable effects on the incidence of PD-associated peritonitis [16]. This would mean that use of biocompatible solutions was not associated with the declining incidence of PD-associated peritonitis noted in this study. This result supports recent results described by Srivastava et al. [30].

Reduced residual renal function is presumed to increase the risk of PD-associated peritonitis. Actually, Han et al. reported reduced residual renal function as a risk factor for peritonitis in patients receiving continuous ambulatory PD [8]. Serum albumin level, which might reflect nutritional status, was significantly higher in patients with residual GFR >5 ml/min/1.73 $m^2$ compared to those with GFR <5 ml/min/1.73 $m^2$. Lower albumin level would thus contribute to the development of peritonitis.

The present study has several limitations. First, this was an observational study based on retrospective data. Data such as laboratory results and treatment of DM were unavailable. For example, we could not assess the effect of serum albumin, a well-known contributing factor for the development of PD-associated peritonitis [31]. We thus could not adjust for confounding factors such as residual renal function, presence of pre-dialysis care, or glycemic control in the Cox analysis. Second, not all patients underwent renal biopsy to determine the diagnosis of renal disease. We cannot exclude the possibility that some patients could have been misclassified. In addition, our study showed selection bias, in that we intentionally selected PD patients with self-management skill. The primary strength of the study was the large number of peritonitis episodes, data on which were collected during 33 years of PD practice.

## Conclusion

The increased risk of peritonitis in diabetics observed in previous periods appears to have disappeared in recent years. Diabetic patients with end-stage renal stage therefore need not avoid selecting the PD modality.

## Acknowledgments

The authors gratefully acknowledge the support and participation of the patients in this study.

## Author Contributions

**Data curation:** Masatsugu Nakao, Akio Nakashima, Nanae Matsuo, Yudo Tanno, Masato Ikeda.

**Formal analysis:** Yukio Maruyama.

**Methodology:** Izumi Yamamoto, Ichiro Ohkido, Hiroyasu Yamamoto, Keitaro Yokoyama.

**Project administration:** Takashi Yokoo.

**Supervision:** Takashi Yokoo.

**Writing – original draft:** Risa Ueda.

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
