## [Decision Letter · Decision Letter 0]

21 Aug 2019

PONE-D-19-20955

Effect of diabetes on incidence of peritoneal dialysis-associated peritonitis

PLOS ONE

Dear Dr Nakao,

Thank you for submitting your manuscript to PLOS ONE. After careful consideration, we feel that it has merit but does not fully meet PLOS ONE’s publication criteria as it currently stands. Therefore, we invite you to submit a revised version of the manuscript that addresses the points raised during the review process

We would appreciate receiving your revised manuscript by Oct 05 2019 11:59PM. To enhance the reproducibility of your results, we recommend that if applicable you deposit your laboratory protocols in protocols.io, where a protocol can be assigned its own identifier (DOI) such that it can be cited independently in the future. For instructions see: http://journals.plos.org/plosone/s/submission-guidelines#loc-laboratory-protocols

We look forward to receiving your revised manuscript.

Kind regards,

Tatsuo Shimosawa, M.D., Ph.D.

Academic Editor

PLOS ONE

Journal Requirements:

2. ‘In ethics statement in the manuscript and in the online submission form, please provide additional information about the patient records/samples used in your retrospective study. Specifically, please ensure that you have discussed whether all data/samples were fully anonymized before you accessed them and/or whether the IRB or ethics committee waived the requirement for informed consent. If patients provided informed written consent to have data/samples from their medical records used in research, please include this information.

Y.M. and M.I. have received scholarship funds from Baxter International Inc. and Terumo Corporation. Y.T. has received research grants from Baxter International Inc. They had no direct involvement in the design or conduct of the study; collection, management, analysis, or interpretation of the data; or the preparation, review, or approval of the manuscript. All other authors have no conflicts of interest to declare.

Reviewers' comments:

Reviewer's Responses to Questions

**Comments to the Author**

1. Is the manuscript technically sound, and do the data support the conclusions?

Reviewer #1: Yes

Reviewer #2: Yes

2. Has the statistical analysis been performed appropriately and rigorously? 

Reviewer #1: I Don't Know

Reviewer #2: Yes

3. Have the authors made all data underlying the findings in their manuscript fully available?

Reviewer #1: No

Reviewer #2: Yes

4. Is the manuscript presented in an intelligible fashion and written in standard English?

Reviewer #1: Yes

Reviewer #2: Yes

5. Review Comments to the Author

Reviewer #1: This article is interesting because it evaluate the effect of diabetes on PD peritonitis over a period of more than 3 decades.

However there are still some problems to accept for publish.

Major

You divided 373 PD patients into 3 groups, according to era. The observation period in 3 groups is different from 7 to 13 years, Era means Period 1 (single bag system), Period 2 (twin bag system), and Period 3 (biocompatible solution).

Is there any article which uses similar era classification? If your original classification, could you mention the history of these development in PD shortly.

You did not mention about the progress of PD devices such as sterlile system. These devices has been developed mainly to reduce the risk of peritonitis. It is better for you to include the progress of PD devices in era classification.

It seems that we often use device for patients whoare empirically risky, for Pariod 3, is there any differences in device usage rate between DM and non DM patients?

I think it is difficult to explain the result by only biocompatible PD solution.

You mentioned the contribution of icodextrin in discussion, even now some icodextrin are not yet biocompatible neutral solution. If you want to describe the contribution of icodextrin, you should show that you use only biocompatible icodextrin in period 3. Otherwise this phrase should be deleted from the discussion.

The progress of devices seems to have changed the cause of peritonitis from touch contamination to exit and tunnel infection, and intaraperitoneal bacterial translocation, could you show the data on transmission of pathogenic bacteria? If difficult, could you consider only recent period 3 peritonitis whether you can show there was no significant difference between DM and non DM group?

Minor

1) Could you show the approval number of the ethics committee of Jikei University Hospital.

2) Table 1

What about patients who are observed over both period? Is there a total of 373 patients, or is there no overlap?

What does PD duration mean in Table 1?

It is better to show the number of total patient-yaer in each period.

Duration of peritonitis is not shown in Table 1. Is there any difference between DM and non DM in each period?

Especially in period 3, the incidence of peritonitis became not significant in both group, what about the duration of peritonitis?

What about the difference in both group about the quality of peritonitis such as simple peritonitis or intractable peritonitis to continue PD (for example MRSA peritonitis, Tuberculosis peritonitis, NTM)?

3) Figure 2

How do you define with peritonitis free time? How do you evaluate the second or more peritonitis in same patient? Kaplan Meir Curve seems to deal with only 1st peritonitis in each period considering the number of each period.

Reviewer #2: The authors conducted an analysis based on abundant data unique to institutions that have played a precursor role in Japanese peritoneal dialysis therapy, and in recent years there is no disadvantage in the prevalence of PD-related peritonitis even in diabetic patients.

This study provides extremely useful information for the selection of dialysis methods for diabetic patients. However, some comments should be raised as follows.

1 ISPD guidelines recommend that the parameters monitored should include rates of specific organisms, and the antimicrobial susceptibilities of the infecting organisms.

In diabetic patients, refractory peritonitis or relapsing, recurrent, and repeat peritonitis may affect the results of this study, so the infecting organisms and the frequency of refractory peritonitis, or relapsing, recurrent, and repeat peritonitis should be mentioned.

2. Exit-site and catheter-tunnel infections are major predis- posing factors to PD-related peritonitis. It should be mentioned whether there were changes in each period of disconnect systems, teaching PD patients and their caregivers, catheter exit site care, and systemic prophylactic antibiotics administration prior to catheter insertion.

3. Since icodextrin is being discussed, changes in the frequency of use of icodextrin should be presented. If information on dialysis fluid is not available even after 2003 in this study, it may be a substitute for discussion based on changes in the frequency of use at other facilities or throughout Japan.

4. There are no data suggesting improved glycemic control in this study, and the results of this study are not related to discussion.

This is because glycemic control can greatly affect the results of this study.

In fact, from 1980 to the present, antidiabetic drugs that can be used especially in dialysis patients have been able to add α-glycosidase inhibitors, glinides, DPP-4 inhibitors, and GLP-1 analogs one after another since the days of insulin alone, and insulin itself has changed significantly. These are assumed to have dramatically improved glycemic control.

If data on glycemic control or diabetes treatment for all patients cannot be obtained in this study, the authors should discuss within the available periods, or cite papers on the transition of glycemic control in diabetic patients during PD. At least the transition of treatments for diabetes should be mentioned.

6. PLOS authors have the option to publish the peer review history of their article (what does this mean?). If published, this will include your full peer review and any attached files.

Reviewer #1: No

Reviewer #2: No

---

## [Author Response · Author response to Decision Letter 0]

11 Oct 2019

We are re-submitting the following manuscript for your consideration: “Effect of diabetes on incidence of peritoneal dialysis-associated peritonitis” (manuscript ID PONE-D-19-20955).

The comments from the three reviewers were extremely helpful. We have worked through each of the comments with the aim of improving the manuscript. My co-authors have all contributed to this revised manuscript and have agreed to this re-submission.

We look forward to hearing from you at your earliest convenience and are very grateful for your consideration.

---

## [Decision Letter · Decision Letter 1]

4 Nov 2019

Effect of diabetes on incidence of peritoneal dialysis-associated peritonitis

PONE-D-19-20955R1

Dear Dr. Nakao,

We are pleased to inform you that your manuscript has been judged scientifically suitable for publication and will be formally accepted for publication once it complies with all outstanding technical requirements.

With kind regards,

Tatsuo Shimosawa, M.D., Ph.D.

Academic Editor

PLOS ONE

Additional Editor Comments (optional):

Reviewers' comments:

Reviewer's Responses to Questions

**Comments to the Author**

1. If the authors have adequately addressed your comments raised in a previous round of review and you feel that this manuscript is now acceptable for publication, you may indicate that here to bypass the “Comments to the Author” section, enter your conflict of interest statement in the “Confidential to Editor” section, and submit your "Accept" recommendation.

Reviewer #1: All comments have been addressed

Reviewer #2: All comments have been addressed

2. Is the manuscript technically sound, and do the data support the conclusions?

Reviewer #1: Yes

Reviewer #2: Yes

3. Has the statistical analysis been performed appropriately and rigorously? 

Reviewer #1: Yes

Reviewer #2: Yes

4. Have the authors made all data underlying the findings in their manuscript fully available?

Reviewer #1: Yes

Reviewer #2: Yes

5. Is the manuscript presented in an intelligible fashion and written in standard English?

Reviewer #1: Yes

Reviewer #2: Yes

6. Review Comments to the Author

Reviewer #1: The authors responds appropriately to Reviewer's comments and the article has been corrected as far as possible.

Reviewer #2: The authors conducted an analysis based on abundant data unique to institutions that have played a precursor role in Japanese peritoneal dialysis therapy, and found a evidence to break past concepts.

The response to the review was also appropriate and I feel that this manuscript is now acceptable for publication.

7. PLOS authors have the option to publish the peer review history of their article (what does this mean?). If published, this will include your full peer review and any attached files.

Reviewer #1: Yes: Shinya Kawamoto

Reviewer #2: No

---

## [Editor Report · Acceptance letter]

4 Dec 2019

PONE-D-19-20955R1 

Effect of diabetes on incidence of peritoneal dialysis-associated peritonitis 

Dear Dr. Nakao:

I am pleased to inform you that your manuscript has been deemed suitable for publication in PLOS ONE. Congratulations! Your manuscript is now with our production department. 

With kind regards,

on behalf of

Prof. Tatsuo Shimosawa 

Academic Editor

PLOS ONE